# Bone Biopsy for Histomorphometry in Chronic Kidney Disease (CKD): State-of-the-Art and New Perspectives

**DOI:** 10.3390/jcm10194617

**Published:** 2021-10-08

**Authors:** Luca Dalle Carbonare, Maria Teresa Valenti, Sandro Giannini, Maurizio Gallieni, Francesca Stefani, Roberto Ciresa, Cristina Politi, Maria Fusaro

**Affiliations:** 1Section of Internal Medicine, Department of Medicine, University of Verona, 37134 Verona, Italy; mariateresa.valenti@univr.it (M.T.V.); stefani.francesca.a@gmail.com (F.S.); roberto.ciresa.rc@gmail.com (R.C.); 2Clinica Medica 1, Department of Medicine, University of Padova, 35128 Padova, Italy; sandro.giannini@unipd.it; 3L. Sacco’ Department of Biomedical and Clinical Sciences, Università degli Studi di Milano “La Statale”, 20157 Milano, Italy; maurizio.gallieni@unimi.it; 4Nephrology and Dialysis Unit, “L. Sacco” Hospital, ASST Fatebenefratelli-Sacco, 20157 Milano, Italy; 5CNR-IFC of Reggio Calabria, Section of Clinical Epidemiology and Biostatistics, 89124 Reggio Calabria, Italy; politicristina89@gmail.com; 6National Research Council (CNR), Institute of Clinical Physiology (IFC), 56124 Pisa, Italy; dante.lucia11@gmail.com; 7Department of Medicine, University of Padova, 35128 Padova, Italy

**Keywords:** bone biopsy, histomorphometry, kidney, osteodystrophy, ROD

## Abstract

The use of bone biopsy for histomorphometric analysis is a quantitative histological examination aimed at obtaining quantitative information on bone remodeling, structure and microarchitecture. The labeling with tetracycline before the procedure also allows for a dynamic analysis of the osteoblastic activity and mineralization process. In the nephrological setting, bone biopsy is indicated to confirm the diagnosis of subclinical or focal osteomalacia and to characterize the different forms of renal osteodystrophy (ROD). Even if bone biopsy is the gold standard for the diagnosis and specific classification of ROD, the use of this approach is very limited. The main reasons for this are the lack of widespread expertise in performing or interpreting bone biopsy results and the cost, invasiveness and potential pain associated with the procedure. In this regard, the sedation, in addition to local anesthesia routinely applied in Italian protocol, significantly reduces pain and ameliorates the pain perception of patients. Concerning the lack of widespread expertise, in Italy a Hub/Spokes model is proposed to standardize the analyses, optimizing the approach to CKD patients and reducing the costs of the procedure. In addition, new tools offer the possibility to evaluate the osteogenic potential or the ability to form bone under normal and pathological conditions, analyzing mesenchymal stem cells and their ability to differentiate in the osteogenic lineage. In the same way, circulating microRNAs are suggested as a tool for exploring osteogenic potential. The combination of different diagnostic approaches and the optimization of the bioptic procedure represent a concrete solution to spread the use of bone biopsy and optimize CKD patient management.

## 1. Introduction

The use of bone biopsy for histomorphometric analysis is a quantitative histological examination aimed at obtaining information on bone remodeling, structure and microarchitecture [1] (Figure 1).

Histomorphometric analyses are classified into structural, microarchitectural and remodeling parameters, with the latter subdivided into static and dynamic categories. The labeling with tetracycline before the procedure also allows a dynamic analysis of osteoblastic activity and mineralization processes [2] (Figure 2). Metabolic bone disorders show different histomorphometric patterns and this approach characterizes and confirms the diagnosis of different metabolic diseases such as sub-clinical or focal osteomalacia, hyperparathyroidism [3], or other endocrine diseases [4], osteoporosis and different forms of renal osteodystrophy (ROD) [5,6]. From another point of view, bone histomorphometry analyses are indicated to clarify cases of unexplained fragility fractures or to evaluate the effects of different treatments on bone, such as antiresorptive or anabolic drugs and their potential side effects [7], as well as the activity of new drugs used for the treatment of metabolic bone diseases [8].

Bone and mineral metabolism alterations are common in chronic kidney disease (CKD) and they are part of a complex systemic osteoporotic condition affecting bone and vascular health, defined by Kidney Disease Improving Global Outcomes (KDIGO) guidelines as Chronic Kidney Disease Mineral and Bone Disorder (CKD-MBD). Until today, bone histomorphometry is the gold standard for the diagnosis of ROD during CKD, providing information that is not available by any other diagnostic approach [9].

In recent decades the definition, classification and treatment of ROD were changed as a consequence of the introduction of new drugs and the prolonged survival of patients (Table 1 [4,10,11,12,13,14,15,16,17,18,19,20,21,22,23]). Indeed, Malluche et al. highlighted [18], in 630 bone biopsies carried out in CKD patients, both in Europe and the USA, from 2003 to 2008, a switch from a high bone turnover, as revealed in the past studies, to 58% of patients with a low bone turnover. These findings could be correlated with the introduction, in 2004, of drugs such as calcimimetics, resulting in excessive bone suppression. In fact, contrasting data were recently published by the Brazilian electronic database (REBRABO) [12] collected from 2015 to 2018, in which the authors found that 50% of the 260 patients with CKD 3-5D had Osteitis Fibrosa. Most likely, among the causes of these results, as specified by the authors, it is necessary to keep in mind the availability of drugs used for the treatment of CKD-MBD [12].

In addition, CKD patients with one or more fragility fractures are challenging to treat, firstly, due to the need to determine whether a patient is affected by osteoporosis and/or the various forms of metabolic bone diseases and, secondly, to choose the best therapeutic approach. The information regarding the actual prevalence of different types of ROD is conflicting. These discrepancies can be attributed to many factors, such as differences in the epidemiological and clinical characteristics of the CKD patients (such as age or ethnicity), the availability of drugs used for the treatment of CKD-MBD and different therapeutic strategies.

For this reason, it is of great importance to have reliable and reproducible diagnostic tools aimed at adequately addressing these complications.

## 2. ROD and Bone Markers

ROD is characterized by abnormalities of calcium, phosphate, parathyroid hormone (PTH), vitamin D metabolism, bone turnover, mineralization, volume or strength, as well as by the presence of vascular or soft tissue calcifications.

The evaluation of bone metabolism, and in particular bone turnover markers, is a useful tool in the management of metabolic bone diseases, but in ROD is not always sufficiently accurate, particularly in presence of low turnover bone diseases for which formation markers, such as bona alkaline phosphatase, are not predictive [24].

Recently, the European Society for Pediatric Nephrology, CKD-MBD and Dialysis and CKD-MBD working groups of the ERA-EDTA stated that, in the pediatric population, the currently available non-invasive measures, including biomarkers of bone, were affected by growth and pubertal status and had a limited sensitivity and specificity in predicting changes in bone turnover and mineralization [25]. This was an important limitation in the management of pediatric patients with CKD.

In summary, we conclude that the evaluation of bone turnover markers, particularly bone alkaline phosphatase can be useful when marked alterations are observed [26].

**Key message:** In borderline cases, the accuracy of markers is low and inadequate for the management of CKD patients.

## 3. ROD and PTH

The parathyroid hormone (PTH) was considered a pivotal marker in the diagnosis and management of ROD. As for turnover markers, the role of this parameter in CKD was recently reconsidered, taking into account its low accuracy in low-turnover renal bone disease [27,28]. 

In the past, a desired PTH level between 150 and 300 pg/mL was suggested for dialysis patients. However, bone biopsy studies showed that a low bone turnover can be seen within this range and sometimes even with higher PTH levels. In a recent study, according to the K/DOQI PTH ranges, 15 patients out of 40 (40%) with iPTH > 300 pg/mL showed an histomorphometric pattern of low turnover [13,14]. Accordingly, the 2017 KDIGO guidelines suggested that the measurements of serum PTH should be coupled to bone-specific alkaline phosphatase. Together they could be used to evaluate bone disease, but only markedly high or low values predicted the underlying bone turnover [29]. In addition, the same guidelines indicated that, in patients with CKD stages 3a, 3b, 4, and 5 not on dialysis, the optimal PTH level was not known.

**Key message:** PTH had a poor predictive power in the low-turnover disease in CKD.

## 4. ROD and Osteoporosis

Dual X-rays absorptiometry (DXA) is considered the gold standard for the diagnosis of osteoporosis and it is one of the main predictors of fragility fractures [28]. The impossibility of measuring trabecular and cortical BMD separately and the interference of aortic calcifications in the measurement of vertebral BMD limit the possibility to adequately classify bone disease in CKD in adults and in adolescents [30], as well as to predict fractures, even using ultrasound techniques [31,32]. 

The Word Health Organization Osteoporosis defines osteoporosis as a reduction in bone density expressed as a T-score lower than −2.5 Standard Deviation (SD) compared to normal population (at the peak of bone mass). This diagnosis can be made independently of bone mass in the presence of fragility fractures. Indeed, bone strength is the result of a combination of bone density and bone quality, consisting of bone microarchitecture, turnover and mineralization. The disorders in bone quality can explain the finding that about half of all osteoporotic fractures occur in patients with T scores > −2.5 SD. However, ROD is characterized by an alteration of bone quality regardless of bone loss [33].

**Key message:** The evaluation of bone density in CKD patients is not always predictive of bone fragility [26]. In fact, DXA provides a two-dimensional evaluation of a three-dimensional structure and offers a very poor spatial resolution [34].

## 5. ROD and Non-Invasive Diagnostic Approach

Different non-invasive approaches were proposed to optimize the management of patients with ROD and different grades of bone fragility [35]. Indeed, the unclear efficacy in predicting fractures of DXA and its inability to specify types of renal osteodystrophy encourages scientific research to identify other non-invasive diagnostic strategies to manage CKD patients. The additional, novel applications for DXA analysis such as Trabecular Bone Score (TBS,) or other approaches such as high-resolution imaging tools, for example peripheral quantitative computed tomography (HR-pQCT), are used to evaluate the effects of kidney disease on cortical, trabecular microarchitecture and bone strength and to adequately predict the risk of fracture [36]. The data are encouraging but it is still debated whether the addition of pQCT or high-resolution pQCT parameters to DXA effectively improves fracture discrimination [37]. However, pQCT has a nominal spatial resolution of about 100 mc [38] and a good signal-to-noise ratio, while bone histomorphometry offers a spatial resolution around the micron and represents, from this perspective, a much more accurate method for evaluating microarchitecture [1].

A combination of different approaches were also proposed to ameliorate the prognostic ability [39]. In this study the authors found that the combination of QCT, DXA and serum biochemical parameters such as PTH, sclerostin, and TRAP-5b (even if not still routine evaluations), were independent predictors of bone loss in patients with CKD-5D and ameliorated the prognostic power of clinical evaluations in these patients.

**Key message**: In general, none of non-invasive approaches demonstrated an adequate diagnostic accuracy for CKD [40].

## 6. ROD and Bone Biopsy

The use of bone biopsy for histomorphometric analysis is the only technique providing the following specific information: (1) the study of bone at the three levels: cells, tissue and Basic Multicellular Units (BMU, Figure 3); (2) the direct evaluation of the turnover at the trabecular level; (3) the static and dynamic evaluation of bone turnover using double tetracycline labelling (Figure 2); and (4) the evaluation of microarchitecture through direct and indirect parameters (Figure 4).

The diagnosis and management of ROD is an important clinical challenge. For this reason, the tetracycline, double-labelled, transiliac bone biopsy for histology and a histomorphometric analysis remains the best clinical approach to describe the static and dynamic parameters of bone turnover and to also evaluate the other qualitative aspects of renal osteodystrophy [41].

Even though this approach was the gold standard for the diagnosis of ROD, a recent European survey showed that bone biopsies were performed rather exceptionally, both for clinical and research purposes. In particular, the clinical research concerning ROD was threatened by the limited availability of technical, clinical and pathological expertise, small patient cohorts, and scientific isolation [42].

## 7. Indications of Bone Biopsy in CKD

Several years ago, the Kidney Disease Improving Global Outcomes (KDIGO) revised the indications of bone biopsy underlying the importance of this approach for the correct classification and the adequate treatment [43].

Recently, the same group revised the guidelines of CKD compared to those of 2009, specifying that, in patients with CKD G3a–G5D, it was reasonable to perform a bone biopsy if knowledge of the type of renal osteodystrophy would impact treatment decisions [42].

On the basis of these most recent findings and consensus statements, we summarized the main indications for bone biopsy in Table 2 [42].

## 8. Classification of ROD by Histomorphometry

Based on the different histomorphometric parameters, the bone disease during CKD was classified taking into account three compartments: turnover, mineralization and volume [41]. The turnover reflected the rate of skeletal remodelling, mineralization described the effectiveness of bone collagen becoming calcified during the formation phase of skeletal remodelling and volume indicated the amount of bone per unit volume of tissue. By using this approach, we could characterize different types of ROD as follows: adynamic bone disease and osteomalacia (low turnover diseases) and osteitis fibrosa and mixed uremic osteodystrophy (high turnover diseases), independently from bone turnover markers or PTH (Figure 5).

## 9. Limitations of Bone Biopsy

Even if, at present, bone biopsy remains the gold standard for the diagnosis and specific classification of renal osteodystrophy (ROD), the use of this approach is very limited. The main reasons for the infrequent use of bone biopsy, other than the lack of widespread expertise in performing or interpreting bone biopsy results and the high costs, are the invasiveness and potential pain associated with the procedure. In this regard, various solutions are proposed, e.g., the use of fluoroscopy for a more precise localization of the biopsy site and the standardization of the intervention [10] or the use of a smaller trephine providing a halved biopsy [12].

In our opinion, the first approach can ameliorate the issues with bone biopsy, but the availability of interventional radiologists is limited and this would hamper the wider application of the method. For the second proposal, according to the previous authors, we believe that smaller samples may result in the loss of information, particularly in terms of turnover, but can be useful for monitoring the treatment or in the follow-up of patients already diagnosed and characterized through a standard biopsy [43].

From another point of view, the use of sedation, in addition to local anesthesia, routinely applied in Italian protocol, significantly reduces pain and ameliorates the negative perceptions of patients.

Concerning the lack of widespread expertise in performing and interpreting bone biopsy, with the support of the Italian Scientific Society of Osteoporosis and Mineral Metabolism Diseases of the Skeleton and of the Italian Nephrology Society, a Hub/Spokes model is proposed to centralize and standardize the analyses, as well as to facilitate the shipment of the bioptic samples to the two hubs, Rome and Verona, creating a national network and optimizing the approach to CKD patients and reducing the costs of the analyses.

## 10. New Perspectives

In addition to the histomorphometric evaluation of bone samples, it is important to know the osteogenic potential of a specific subject or the ability of bone to renew itself under normal and pathological conditions, particularly in CKD.

In recent years, the studies aimed at evaluating mesenchymal stem cells, as well as studies conducted to evaluate molecular markers involved in osteogenic differentiation, ensured the understanding of osteogenic potential in a given physiological or pathological context.

Mesenchymal stem cells (MSCs) are undifferentiated cells able to differentiate to different cellular lineages such as osteoblastic, chondrocyte or adipocyte cells. In particular, specific cellular signals can induce the gene expression of the osteoblastic master gene, RUNX2, which, by activating specific downstream genes, such as the transcription factor SP7, directs MSCs in an osteogenic sense [44]. In osteoporotic patients, an increased number of circulating MSCs associated with a reduced expression of RUNX2, SP7 and osteoblastic maturation-associated genes suggests an impaired osteogenic potential of these patients [45]. Therefore, the molecular profile associated with mesenchymal cells reflects the osteogenic potential in an individual. Importantly, circulating MSCs are non-invasive biomarkers as they can be obtained from a peripheral blood sample [46,47].

Epigenetic alteration, such as the methylation and acetylation of DNA and histones is involved in the aging process. It was demonstrated that KAT6A, an enzyme responsible for histone acetylation, acted as a transcriptional coactivator of RUNX2 and that the alterations of KAT6 impaired skeleton genesis [48]. Recently, it was demonstrated that KAT6A regulated the aging of MSCs and counteracted ROS accumulation through the Nrf2/ARE signaling pathway [48]. Thus, a lower KAT6 expression reduced the osteogenic potential of the MSCs and the investigations of this marker in circulating MSCs allowed knowledge of the osteogenic potential. In addition, circulating microRNAs (miR), which were small, non-coding molecules, were suggested as a tool for exploring osteogenic potential. miR223, miR148a, miR31, miR21-5p and miR503 were just some of the markers associated with osteoclastic activity while miR322, miR181, miR143, miR 182 and miR155 were involved in osteoblastic activity [49]. Thus, an individual, circulating microRNA profile could reflect the osteogenic potential. In addition, the long and non-coding RNAs (lncRNAs), i.e., RNA molecules with a length of > 200 nucleotides, regulated gene expression in different cellular processes and it was observed that lncRNAs were associated with osteogenesis and bone diseases [50,51,52]. The lncRNA targeting DANCR was observed in human circulating monocytes [53]. As DANCR correlated to IL6 and TNFα in patients affected by osteoporosis it was suggested that the long and non-coding RNA-DANCR could be used as a biomarker in osteoporosis [53]. The implementation of bone biopsy with such non-invasive approaches for the study of bone osteogenic precursors could further improve the diagnostic accuracy in this specific setting.

## 11. Conclusions

The use of bone biopsy remains the gold standard in the diagnosis of metabolic bone diseases, particularly in CKD patients.

The Italian bone biopsy protocol, as well as the built network based on the hub/spoke model, can ameliorate the appeal of this procedure, standardize the results, and create a national database optimizing the management of patients with metabolic bone diseases.

## 12. Highlights

Bone and mineral metabolism alterations are common in chronic kidney disease (CKD);In borderline cases, the accuracy of markers are low and inadequate for the management of CKD patients;PTH has a poor predictive power in low turnover disease in CKD;In CKD patients, the predictive powers of Bone Densitometryand non-invasive approaches are low because they provide a two-dimensional evaluation of a three-dimensional structure, and thus offer a very poor spatial resolution;The use of bone biopsy is the gold standard for the diagnosis and specific classification of Renal Osteodystrophy (ROD);The Italian biopsy protocol, as well as the built network based on the hub/spoke model, can ameliorate the appeal of this procedure.

## Figures and Tables

**Figure 1 jcm-10-04617-f001:**
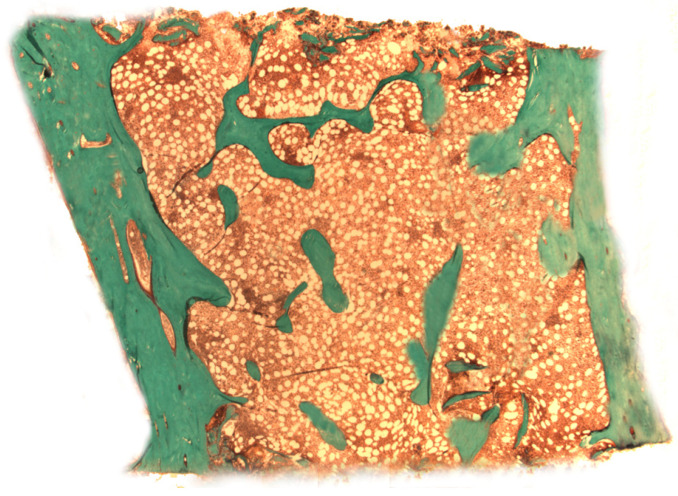
Representation of a Goldner trichrome-stained bone section. In green: the mineralized matrix, in red: the osteoid.

**Figure 2 jcm-10-04617-f002:**
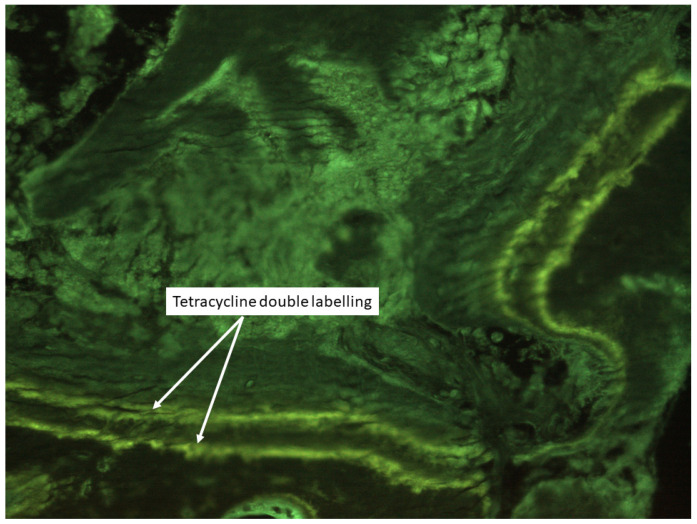
Example of double tetracycline labelling. The two green lines represent the mineralization surfaces while the Mineral Apposition Rate is the distance between the two lines expressed in microns.

**Figure 3 jcm-10-04617-f003:**
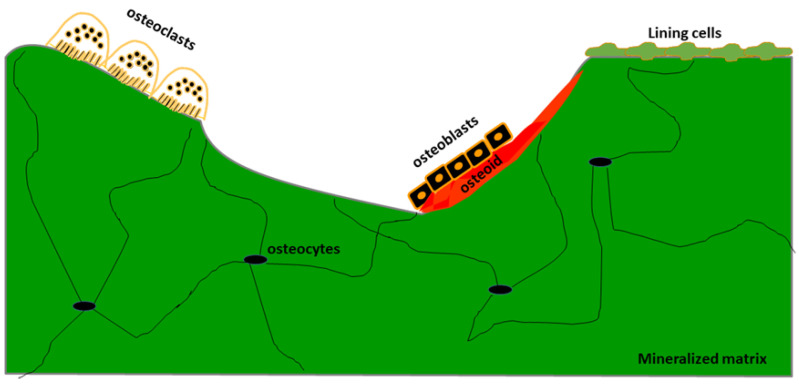
Schematic representation of Bone Multicellular Unit (BMU).

**Figure 4 jcm-10-04617-f004:**
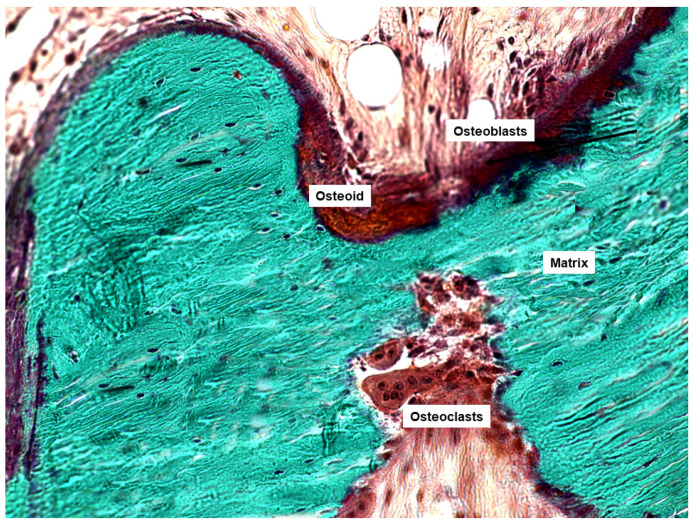
Goldner trichrome-stained bone section in which we can see osteoblasts, osteoclasts and osteoid, the main product of osteoblasts.

**Figure 5 jcm-10-04617-f005:**
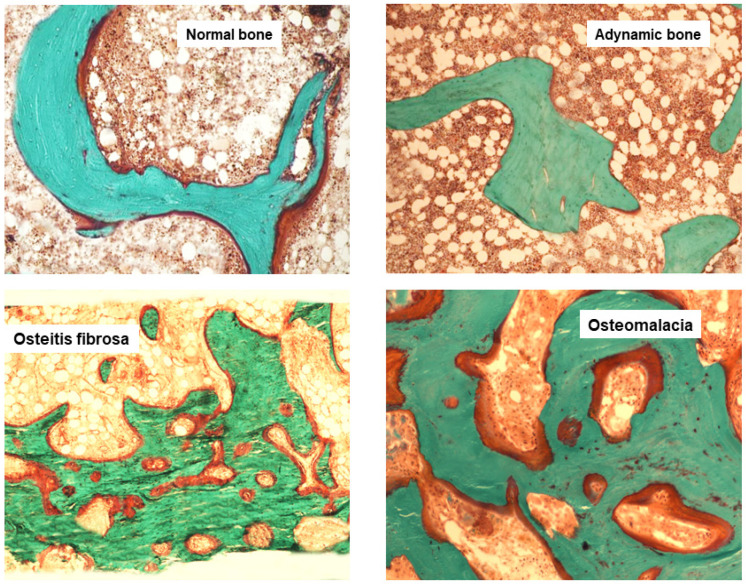
Goldner trichrome-stained bone section representing the main histomorphometric patterns of Renal Osteodystrophy ROD.

**Table 1 jcm-10-04617-t001:** The main literature on bone biopsy in recent decades.

Authors	Design of the Study	Number of Patients	Type of Bone Disease	Journal, Year
Aaltonen et al.	Observational	*n* = 26 ESRD	61% low, 27% normal, 12% high bone turnover	Calcif Tissue Int. 2021
Lavigne et al.	Observational	*n* = 11 CKD	45.4% adynamic bone disease 18.1% high 18.1% mixed renal osteodystrophy 9.1% osteomalacia 9.1% not defined	J Nephrol. 2021
Salam et al.	Cross-sectional	*n* = 43 CKD stages IV-V	40% high, 34% normal, 26% low bone turnover	Bone, 2021
Carbonara et al.	Observational	*n* = 260 CKD stages III-V	High bone turnover	J. Bras. Nefrol. 2020
Novel-Catin et al.	Observational	*n* = 68, ESRD	45% Osteitis fibrosa 21% Mixed uremic osteodystrophy 12% Adynamic bone disease 10% Osteomalacia	Bone, 2020
Liangos et al.	Observational	33, CKD stage III-V 53, on hemodialysis	High-turnover mixed uremic osteodystrophy	Kidney Blood Press Res. 2018
Sharma et al.	Observational	14, CKD stage V	50% high, 29% normal, 21% low bone turnover	Am J Nephrol, 2018
Evenepoel et al.	Observational	36, kidney transplant	44.4% low, 52.8% normal, 2.8% high bone turnover	Kidney Int., 2017
Sprague et al.	Cross-sectional	492, on hemodialysis	59% low, 24% normal, 17% high bone turnover	Am J Kidney Dis., 2016
Malluche et al.	Observational	630, CKD stage V	58% low, 24% high, 18% normal bone turnover	JBMR 2011
Lehmann et al.	Observational	36, CKD stages III-IV 92, CKD stage V	47.2% Osteitis fibrosa 61.4% Osteitis fibrosa	Clin Nephrol, 2008
Miller et al.	Observational	6, CKD stages IV-V	33% low, 33% high bone turnover, 33% osteomalacia	CJASN 2008
Coen et al.	Observational	79, CKD 107, on hemodialysis	69% mixed osteodystrophy 11% adynamic bone disease 2.5% hyperparathyroidism 1% osteomalacia 57% hyperparathyroidism 28% mixed osteodystrophy 11% adynamic bone disease 2.8% osteomalacia	Nephron, 2002
Gerakis et al.	Observational	62, on hemodialysis	64.5% hyperparathyroidism 22.6% adynamic bone disease 9.7% mixed bone disease 3.2% osteomalacia	J Nephrol., 2000
Jørgensen et al.	Observational	205, kidney transplan	24% low, 60% normal, 16% high turnover	Bone, 2021

ESRD: End Stage Renal Disease; CKD: Chronic Kidney Disease.

**Table 2 jcm-10-04617-t002:** Synthetic updated indications for bone biopsy in CKD.

(1) In patients with CKD G3a–G5D, it is reasonable to perform a bone biopsy if knowledge of the type of renal osteodystrophy will impact treatment decisions (evidence not graded)
(2) In patients with CKD G3a–G5D with biochemical abnormalities of CKD-MBD and low BMD and/or fragility fractures, we suggest that treatment choices take into account the magnitude and reversibility of the biochemical abnormalities and the progression of CKD, with consideration of a bone biopsy (evidence 2D).
(3) During the first 12 months after kidney transplant, in patients in with an estimated glomerular filtration rate greater than approximately 30 mL/min/1.73 m^2^ and a low BMD, it is reasonable to consider a bone biopsy to guide treatment (evidence not graded).

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
