# Peer review of "Bone Biopsy for Histomorphometry in Chronic Kidney Disease (CKD): State-of-the-Art and New Perspectives"

_jcm, 2021, doi:10.3390/jcm10194617_

Round 1

Reviewer 1 Report

1)  In the introduction section on page 2, combine the concepts of line 3-4
with line 3-6 on page 3

2) Add a legend with abbreviations

3) Page 4 from line 3 to line 6 is a repetition

4) In the paragraph ROD and PTH he explains the concept better by expanding it

5) In the paragraph ROD and osteoporosis add bibliography in the last concept

6) You could combine the paragraph ROD and osteoporosis with bone densitometry and ROD

Author Response

Reviewer 1

1)  In the introduction section on page 2, combine the concepts of line 3-4

with line 3-6 on page 3

We combined the concepts, as suggested

2) Add a legend with abbreviations

As suggested, we added a list of abbreviations.

3) Page 4 from line 3 to line 6 is a repetition

We corrected and harmonized the text.

4) In the paragraph ROD and PTH he explains the concept better by expanding it

As suggested, we expanded the concept for a better understanding.

5) In the paragraph ROD and osteoporosis add bibliography in the last concept

As suggested, we added a reference in the last concept of the paragraph.

6) You could combine the paragraph ROD and osteoporosis with bone densitometry and ROD

As suggested, we combined the two paragraph on osteoporosis and bone density.

Reviewer 2 Report

In this article the authors highlighted the role of bone biopsy for histomorphometric analysis to confirm the diagnosis of subclinical or focal osteomalacia and to characterize the different forms of renal osteodystrophy. Overall this is a well written review article. However, some queries need to be clarified before publication.

  1. Table 1 lists 15 references regarding the main literature on bone biopsy of the last decades. Please discuss more about the changes of the definition, classification and treatment of renal osteodystrophy.
  2. In the section of "ROD and non-invasive diagnostic approach", the authors that a combination of different approaches has also been proposed to ameliorate the prognostic ability (39). More detailed information about the study will be favored.    
  3. It will be better to provide each type of ROD in Figure 5. 

Author Response

Reviewer 2

1.Table 1 lists 15 references regarding the main literature on bone biopsy of the last decades. Please discuss more about the changes of the definition, classification and treatment of renal osteodystrophy.

As suggested, we expanded the paragraph about the changes of the definition, classification and treatment of renal osteodystrophy.

2.In the section of "ROD and non-invasive diagnostic approach", the authors that a combination of different approaches has also been proposed to ameliorate the prognostic ability (39). More detailed information about the study will be favored.  

As suggested, we added more detailed information of this study in the text.

3.It will be better to provide each type of ROD in Figure 5.

Thank you for this suggestion: we changed the figure providing each histomorphometric type of ROD.